# Longitudinal 3D Study of Anterior Tooth Wear from Adolescence to Adulthood in Modern Humans

**DOI:** 10.3390/biology10070660

**Published:** 2021-07-13

**Authors:** Nikolaos Gkantidis, Konstantinos Dritsas, Meret Gebistorf, Demetrios Halazonetis, Yijin Ren, Christos Katsaros

**Affiliations:** 1Department of Orthodontics and Dentofacial Orthopedics, University of Bern, CH-3010 Bern, Switzerland; dritsaskonstantinos@gmail.com (K.D.); meret@gebi.ch (M.G.); christos.katsaros@zmk.unibe.ch (C.K.); 2Department of Orthodontics, W.J. Kolff Institute, University Medical Center Groningen, University of Groningen, 9700RB Groningen, The Netherlands; y.ren@umcg.nl; 3Department of Orthodontics, School of Dentistry, National and Kapodistrian University of Athens, GR-11527 Athens, Greece; dhalaz@dent.uoa.gr

**Keywords:** tooth wear, anterior teeth, modern humans, quantitative assessment, three-dimensional imaging, long-term outcome

## Abstract

**Simple Summary:**

Tooth wear is the loss of tooth substance during everyday functioning by means other than dental caries. It is expected at a certain level in every person, and it increases with age. In the last decades, due to the increased life expectancy and high patient demands, it has become an important problem modern dentistry has to face. However, the average amount of tooth wear among individuals remains controversial. The purpose of this clinical study was to precisely detect the extent of anterior tooth wear over a thirteen-year period, from adolescence to adulthood. The present study revealed the very high tooth wear occurrence in the population already at early adulthood and showed that wear monitoring at an individual level is important for dentists to enable the better understanding of the problem and allow timely targeted interventions for patients in need. These might be preventive, such as the prescription of mouth guards to limit grinding and the cessation of harmful habits that lead to tooth wear, or interceptive, such as the restoration of the lost tooth substance to stop the progress of the condition and improve esthetics and function.

**Abstract:**

In modern humans, tooth wear can easily be observed as a loss of tooth substance, but its precise measurement is problematic. The aim of this longitudinal cohort study was to determine the precise amount of occlusal tooth wear in the anterior permanent dentition from adolescence to adulthood. Corresponding tooth crowns from serial 3D digital dental models of 72 individuals were best fit-approximated by applying novel, highly accurate 3D superimposition methods. The superimposed crowns were simultaneously sliced on intact structures, and the differences in the volumes of the subsequent occlusal parts were calculated. Over a thirteen-year period, there was an average loss of anterior occlusal surfaces of 1.58 mm^3^ per tooth. Tooth surface loss in at least one tooth was higher than 1 mm^3^ in 93.1% of the human subjects. Tooth wear severity differed by sex and tooth type, with males showing higher values versus females and upper canines versus other anterior teeth. The study revealed the endemic occurrence of occlusal anterior tooth wear, highlighting the need for monitoring of the condition in the population to identify high-risk patients and enable timely interventions. The novel methods applied here on 3D digital models are recommended for this.

## 1. Introduction

In modern humans, tooth wear is a physiological process that progresses slowly over time [1,2,3]. However, loss of tooth matter may also be atypical for the patient’s age due to pathological conditions associated with attrition, abrasion, and erosion processes [2,4]. These conditions may coexist and act synergistically in varying degrees in individual patients [5].

Occlusal tooth wear is a very common finding in the general population, with anterior teeth exhibiting greater wear [2,6,7,8,9]. The etiology of tooth wear is multifactorial. Males show more often and more severe tooth wear than females [1,2]. Dietary factors, including acidic foods, have also been shown to have a critical role [2,4,6,9,10]. Certain occlusal factors, such as overjet and anterior protected articulation, have been associated with the progression of occlusal wear [11]. Subjects with different malocclusions have demonstrated different tooth wear patterns [12,13]. Masticatory function has also been associated with tooth wear [14].

In regard to the effects of tooth wear on individuals, the severity of occlusal/incisal tooth wear was found to be associated with dentine hypersensitivity severity [10]. However, the most important patient concern regards esthetics, followed by sensitivity, whereas functional problems and pain are less prevalent [15]. A series of studies designated anterior tooth wear as esthetically unpleasant [16,17,18]. Furthermore, anterior tooth wear negatively affected the quality of life in elder nursing home patients [19].

Proper monitoring of tooth wear is crucial for successful management [20]. Erosion has been found to progress more in individuals with previous signs of erosion, emphasizing the importance of an early diagnosis [21]. However, dental patients might be unable to recognize tooth wear and seek treatment, even in severe cases [5]. Additionally, conventional methods applied by dental professionals have been found to show a limited capacity to follow wear progress over time [22,23]. Therefore, novel clinically applicable 3D approaches have been developed for this [24,25,26,27].

The precise amount of tooth wear occurring in the long term on a group of individuals has not yet been fully explored. Longitudinal, quantitative in vivo studies on medium- to long-term tooth wear occurrence are scarce [28,29]. Thus, the purpose of the present study was to assess the amount of occlusal anterior tooth wear evident from adolescent to adulthood over a long-term period. To fulfill this purpose, recently developed 3D superimposition methods [24,25,27] were applied on serial post-orthodontic treatment digital dental models.

## 2. Materials and Methods

This study was reported according to the guidelines provided in the STROBE statement [30].

### 2.1. Ethical Approval

This research protocol was approved by the Research Ethics Committee of the canton of Bern, Switzerland (Project-ID: 2019-00326). All participants signed an informed consent form approving the use of their data for research purposes.

### 2.2. Sample

The sample consisted of individuals that completed their orthodontic treatment in a private practice in Grenchen, Switzerland and were selected for a previous project that tested the performance of fixed orthodontic retainers [31]. Time point 1 (T1) was defined as the point when the participants finished their orthodontic treatment. Time point 2 (T2) was a recall appointment performed to obtain, among others, alginate impressions and intraoral photos. T1 records were taken between 2000 and 2006 and T2 records between 2015 and 2016. A consecutive sample selection was applied to include all patients that fulfilled the eligibility criteria and accepted to participate in the study. The original sample collection was performed using the following inclusion criteria: (1) treated with fixed orthodontic appliances, (2) treated by the same orthodontist, (3) maxillary and mandibular retainers bonded at the anterior teeth immediately after active orthodontic treatment, (4) no orthodontic retreatment, (5) White patients, and (6) non-syndromic individuals. The following additional inclusion criteria were applied in the present study: (1) age at debonding between 12 and 24 years, (2) debonding 9–16 years prior to T2, (3) T1 and T2 casts of good quality in the anterior region of the dental arch, and (4) intact natural teeth without incisal or whole crown reconstructions.

The T1 and T2 dental stone models generated through alginate impressions were scanned using a 3D laboratory surface scanner (stripe light/LED illumination; full dental arch accuracy <20 μm; Laboratory scanner D104a, Cendres + Métaux SA, Biel/Bienne, Switzerland). This scanner resolves distances smaller than 5 μm between corresponding surfaces of repeated single jaw scans [24]. The subsequent maxillary and mandibular 3D surface models, consisting of approximately 600,000–900,000 triangles each, were exported as STL files. These files were used to measure anterior tooth wear, as described below. 

Inadequate quality casts and broken or restored teeth were identified through the visual inspection of intraoral photos and 3D digital dental models. In case of doubt, the medical and dental histories of the patients were reviewed, the corresponding tooth pairs were superimposed, and color-coded distance maps were created for visualisation purposes. Afterwards, a joint decision was made by the first two authors (3 models).

From the 88 individuals included in the previous study [31], data from 72 individuals (54 females and 18 males) were used here. From the 16 excluded patients, 9 did not meet the age criteria, 2 the years from debonding, and 5 had poor-quality casts. Finally, tooth wear was measured on 690 individual teeth (Table 1). The median interval between the two time points was 12.8 years (interquartile range (IQR): 1.4). The median ages at the T1 and T2 points were 14.3 (IQR: 1.8) years and 27.4 (IQR: 2.2) years, respectively. All patients had acceptable occlusion at both time points, with the vast majority having positive overjet and overbites up to 4 mm and Angle Class I. The detailed occlusal characteristics of the sample are provided in Table 2.

### 2.3. Tooth Wear Assessment Workflow

Tooth wear was measured by applying previously developed and validated 3D superimposition techniques [24,25,27] on corresponding individual anterior tooth crowns of the T1 and T2 models. All processing was performed using Viewbox 4 software (version 4.1.0.8 BETA, dHAL Software, Kifissia, Greece). The tooth crowns of the six anterior maxillary and mandibular teeth were selected on the T2 dental models. Each T2 individual tooth crown was superimposed on the corresponding T1 tooth crown using the complete T2 clinical crown as a superimposition reference under the following software settings: 20% estimated overlap of meshes, matching point to plane, exact nearest-neighbor search, 100% point sampling, exclude overhangs, and 50 iterations. The superimposition aimed at the best fit approximation of the two models achieved through the implementation of an iterative closest point algorithm (ICP) [32]. Before applying the final setting, the T1 and T2 crowns were manually approximated using a 100% estimated overlap of the meshes to facilitate the registration process.

The superimposed T1 and T2 crown models were simultaneously sliced using one or more planes. The subsequent occlusal crown parts were transformed into watertight models through a hole-filling process. For each T1–T2 crown pair, the T1 occlusal part volume was subtracted from the T0 volume, providing the tooth wear measurement in mm^3^. The superimposition and measurement workflow are shown in Figure 1a–d and were validated previously [24,25,27].

### 2.4. Method Error

The whole superimposition and measurement process was repeated by the same examiner (K.D.) at least two weeks after the first measurement to assess method error. Twenty-four randomly selected teeth, representing each tooth type twice, were remeasured.

### 2.5. Statistical Analysis

Statistical analysis was performed in IBM SPSS statistics for Windows (Version 26.0. IBM Corp: Armonk, NY, USA). Data normality was tested using the Kolmogorov–Smirnov and Shapiro–Wilk tests and through the visualization of data distribution graphs. There were no significant deviations from normality, and thus, parametric statistics were applied.

Descriptive statistics were performed to present the variables of interest using exact measures and graphical representations.

Differences between the sides of the mouth (right left) were tested separately for each tooth type and jaw using paired *t*-tests. Based on the outcomes of these tests, the average values of the right and left sides were calculated for each tooth type.

Differences in the amount of tooth wear (dependent variable) among tooth types (fixed factor, 6 levels: upper canine, upper lateral incisor, upper central incisor, lower canine, lower lateral incisor, and lower central incisor) and between sexes (fixed factor, 2 levels: male and female) were tested using an analysis of covariance (ANCOVA; general linear model, full factorial). The patient and the duration of the assessment period were set as covariates to account for the matching and clustering effects and correct for the duration of the assessment period, respectively. Age at T1 was not included in the model, based on exploratory tests through bivariate correlations with the different tooth types (Appendix A). Following significant results, parameter estimates were calculated, and pairwise comparisons were performed.

The association of tooth wear between the different tooth types was investigated through bivariate Spearman’s correlations.

The amount of error was assessed through the absolute differences between repeated measurements. The testing of systematic errors and assessments of individual measurements were performed through the Bland–Altman method.

The level of significance was set at an alpha level of 0.05. A Bonferroni correction was applied when multiple pairwise comparisons were performed for similar outcomes. The unit of analysis was each tooth crown.

## 3. Results

### 3.1. Method Error

There were no systematic differences between repeated measurements (*n* = 24; 95% Limits of Agreement: −0.13, 0.25 mm^3^). The average of the differences between repeated measurements was 0.06 mm^3^ on an average tooth wear of 1.68 ± 1.61 mm^3^. The average of the absolute differences between repeated measurements was 0.09 mm^3^, with a standard error of 0.014 mm^3^. The maximum difference detected was 0.24 mm^3^. Thus, the amount of error was considered negligible. The relevant Bland–Altman plot is shown in Figure 2.

### 3.2. Tooth Wear Differences between Contralateral Sides

There were no significant differences between contralateral teeth of the same type (Appendix A). Thus, the wear amounts of each tooth type per side were averaged for further analysis.

### 3.3. Tooth Wear among Tooth Types and between Sexes

The average occlusal wear over the assessment period was 1.58 mm^3^ per anterior tooth. If we consider a cutoff value of 1 mm^3^ as of clinical importance, 50.4% of the tested teeth (348 out of 691) showed higher wear amount. Furthermore, 67 out of the 72 tested individuals (93.1%) had at least one anterior tooth with occlusal wear higher than 1 mm^3^.

The ANCOVA identified a significant effect of tooth type and sex on the tooth wear amount (Table 3 and Figure 3). The parameter estimates indicating the effect of the tested factors on the tooth wear amount (dependent variable) are provided in Appendix A and the estimated marginal means for the factors sex and tooth type in Appendix A. Males showed, on average, 2.66 (95% CI: 2.38–2.95) mm^3^ occlusal tooth wear in the anterior dentition, whereas females showed 1.21 (95% CI: 1.05–1.37) mm^3^. The upper canine presented the highest tooth wear amount (average: 3.30; 95% CI: 2.93–3.67 mm^3^), followed by the upper central incisor (average: 2.27; 95% CI: 1.86–2.68 mm^3^) and the lower canine (average: 2.13; 95% CI: 1.73–2.52 mm^3^). On the contrary, the upper lateral incisor showed the least tooth wear amount (average: 0.90; 95% CI: 0.52–1.29 mm^3^). Although males displayed approximately 2.2 times more tooth wear than females, the pattern of wear severity per tooth type was similar for both sexes (Figure 3). Furthermore, there were moderate-to-strong correlations (average Spearman’s ρ: 0.57; range: 0.39–0.81; *p* < 0.005) between the wear amounts detected on different tooth types of the same individual (Appendix A).

Three-dimensional occlusal wear patterns of representative cases of the average and the maximum amounts detected per tooth type, for the entire sample, are shown in Figure 4.

## 4. Discussion

The common occurrence of anterior tooth wear, in combination with the high esthetic demands and the increasing life expectancy of modern individuals, designates tooth wear an important challenge for contemporary dentistry. However, so far, there is limited information on the extent of the problem in the population over a long-term period. The present study applied highly accurate 3D superimposition techniques on a group of 72 adolescents, followed over a thirteen-year period. The average occlusal wear per anterior permanent tooth was 1.58 mm^3^. Approximately 50% of the tested teeth showed values higher than 1 mm^3^, whereas 93% of the tested individuals had at least one such tooth.

In the literature, we were able to identify only one study quantifying long-term tooth wear, and this was cross-sectional and used 2D radiographs. Τhe average tooth wear of permanent incisors, between 10 and 70 years of age, was 1.01 mm in the maxilla and 1.46 mm in the mandible, measured as the tooth crown length reduction [33]. A longitudinal study identified a crown length reduction of approximately 100–200 μm on the maxillary incisors of 10 middle-aged patients with nocturnal bruxism over a 4-year period [28]. In a previous study by our group, we calculated that 0.5-, 1-, and 2-mm vertical tooth crown loss corresponds to approximately 1.5-, 3.4-, and 8.5-mm^3^ volumetric loss, respectively [25]. Another longitudinal study performed a qualitative assessment of anterior tooth wear in orthodontically treated patients over a 5-year retention period [3]. Finally, a 3-year longitudinal study on 29 adults referred to a dental center for tooth wear management reported an average volume loss of 0.93 mm^3^ for the occlusal surfaces of all teeth, excluding third molars. More specifically, the measured volume loss was 2.53 mm^3^ on molars and 0.83 mm^3^ on upper central incisors [29]. Thus, the present study is advantageous to the available longitudinal studies, since it investigated a larger sample, which was followed over a much longer period and tested through newly developed accurate 3D methods.

The present findings revealed the endemic occurrence of anterior tooth wear, highlighting the need for proper diagnosis, prevention, and management tools. The identification of prognostic factors is crucial to define high-risk individuals for severe tooth wear and facilitate timely management. In accordance with previous findings [1,2], males showed 2.2 times higher tooth wear amounts than females. Interestingly, the pattern of tooth wear severity per tooth type was similar for both sexes. Apart from sex, the fact that tooth wear amounts within a dentition were correlated indicates a general effect of the predisposing or causative factors on the anterior dentition. This is not surprising when considering that the known associated factors, including sex [1,2], diet [2,4,6,9,10], occlusion [11,12,13], and mastication pattern [14], are not expected to have effects solely limited to specific teeth.

However, the multifactorial etiology of tooth wear complicates the accurate identification of high-risk individuals prior to the development of the problem. Additionally, there is evidence supporting that tooth wear progresses faster within a certain time period in individuals with previous wear signs [21]. Probably, such individuals continue to occupy the predisposing and causative factors that led to tooth wear at the earlier stages. Additionally, the fact that wear between tooth types is correlated within a dentition indicates that, in individuals featuring the predisposing factors, a number of teeth are affected. Thus, the detection of tooth wear at the early stages and the proper monitoring of its progression over time is imperative, especially in high-risk patients, to prevent this condition prior to a stage where relatively complex treatments, with questionable medium-to-long-term prognoses, might be required [20].

Apart from the known prognostic factors, such as sex [1,2] or acidic food [4,6,9,10], early diagnosis at an individual level will define the need for a detailed assessment with regular follow-ups. Eventually, this will enable the early identification and elimination of potential causative factors in high-risk patients. The methodology applied here offers a risk-free, low-cost, convenient, and highly accurate tooth wear assessment at a micrometer scale, provided that serial 3D digital dental models are available [24,25,27]. The high incorporation of intraoral 3D scanners in daily dental practice facilitates this purpose [34,35]. Currently, various intraoral scanners are available on the market, enabling the easy acquisition of highly accurate dental models, especially when considering structures of limited extent, such as single teeth [24,25,27,35]. We strongly recommend the acquisition of an intraoral model for every patient that attends a dental practice at the early stages of permanent dentition. This can be used as a reference to accurately monitor the progress of morphological alterations in the oral structure, such as tooth wear [24,25,27], by superimposing them with future models of the same patient.

The present study offers unique 3D information on the amount of tooth wear that should be expected in a young population from adolescence to early adulthood. The use of orthodontically treated individuals minimized the potential confounding from occlusal factors that have been shown to be associated with tooth wear development [11,12,13]. On the other hand, it could restrict the generalizability of the findings, though, in modern societies, it is quite common for patients with malocclusion to receive orthodontic treatment. Furthermore, a large-scale observational study on general practice patients did not identify any association between tooth wear and previous orthodontic treatments [1]. Non-White individuals were excluded from the study, since they represented a small percentage of the patients treated in the place of a sample collection. Thus, few patients with highly variable racial and cultural characteristics could have been included, potentially confounding the outcomes. In the present sample, the studied subjects had clinically acceptable occlusion and did not wear any removable appliances to retain their orthodontic treatment outcome, which might be another confounding factor. On the other hand, confounding deriving from dimensional instability of the digitized orthodontic stone models might be evident [7]. Large distortions, though, were excluded through the visual inspection of intraoral photos and primarily of distance maps of superimposed teeth.

Another limitation of the study is that the T1 data were retrospectively retrieved. Therefore, potential related factors, such as diet or masticatory function, were not assessed. Although specific risk factors on an individual level cannot be assessed, the tested sample can be considered representative of the tested population for reasons discussed earlier. Retrospective data collection might introduce selection and detection bias. Selection bias is not expected in this study, since the original sample was consecutively selected for another purpose—namely, to test the orthodontic retainer performance. To reduce the selection bias, all patients of the original sample who fulfilled the eligibility criteria were included. Detection bias was also not expected, since the applied methods were automated and standardized. Finally, the sample originated from a Central European population, and the outcomes need to be confirmed in other populations.

## 5. Conclusions

The present study revealed the very high occurrence of anterior tooth wear in a population over a thirteen-year period, from adolescence to adulthood. The anterior occlusal surfaces averaged a loss of 1.58 mm^3^ per tooth, with 93.1% of the tested individuals having at least one tooth worn more than 1 mm^3^. Tooth type and sex were significant factors related to tooth wear development, with males and upper canines showing the highest tooth wear amounts. The study revealed the endemic occurrence of occlusal anterior tooth wear, highlighting the need for closer monitoring of the condition in the population. The novel powerful 3D imaging methods applied here can facilitate this purpose, enabling the better understanding of the problem and timely targeted interventions for patients in need.

## Figures and Tables

**Figure 1 biology-10-00660-f001:**
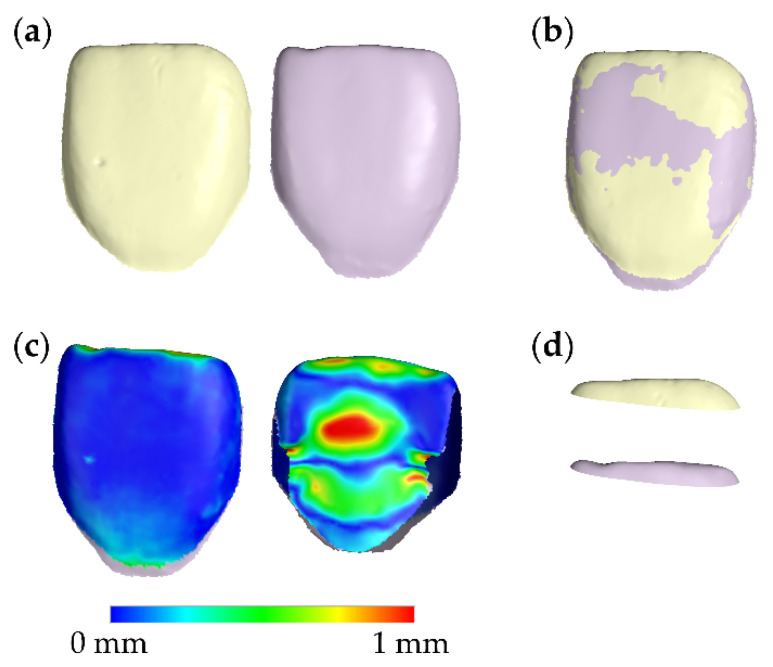
Tooth wear measurement process. (**a**) Corresponding T1 (light yellow) and T2 (light purple) central incisor tooth crowns. (**b**) Best-fit superimposition of the T1 and T2 tooth crowns. (**c**) Buccal and palatal aspects of the color-coded distance map of the superimposed T1 and T2 crowns. (**d**) Corresponding T1 and T2 occlusal crown parts, created following the simultaneous slicing of the superimposed T1 and T2 crowns. The subtraction of these volumes provided the tooth wear measurements used in this study. T1: time point at end of orthodontic treatment. T2: time point on average 13 years after T1.

**Figure 2 biology-10-00660-f002:**
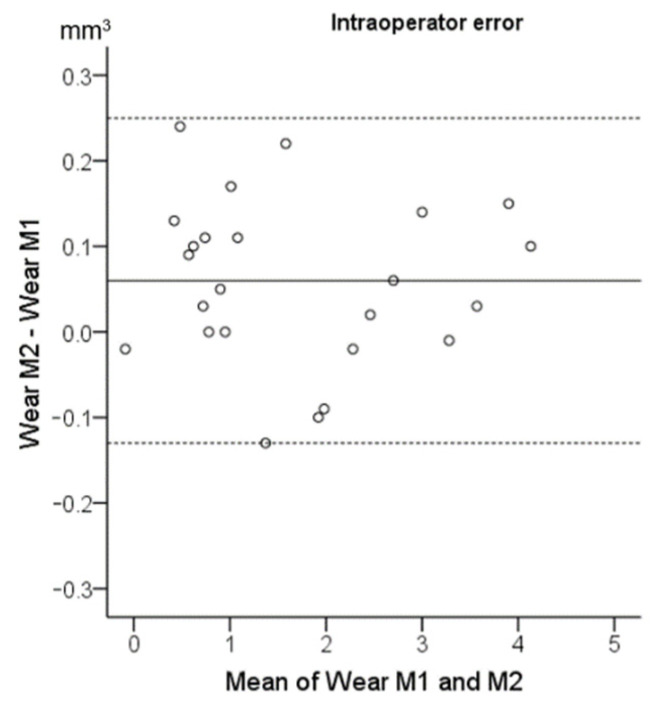
Bland–Altman plot showing the intraoperator reproducibility of the tooth wear assessment in mm^3^. The axes lengths represent the true range of the measured tooth wear values. The continuous horizontal line shows the mean (0.06 mm^3^) and the dashed lines the 95% Limits of Agreement (−0.13, 0.25 mm^3^). M1: first set of measurements. M2: repeated measurements.

**Figure 3 biology-10-00660-f003:**
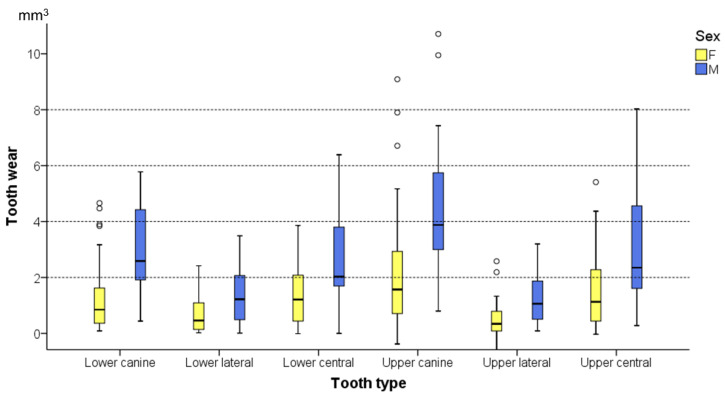
Box plots showing tooth wear for each tooth type and sex. The upper limit of the black line represents the maximum value, the lower limit the minimum value, the box the interquartile range, and the horizontal black line the median value. Outliers are shown as black circles (°).

**Figure 4 biology-10-00660-f004:**
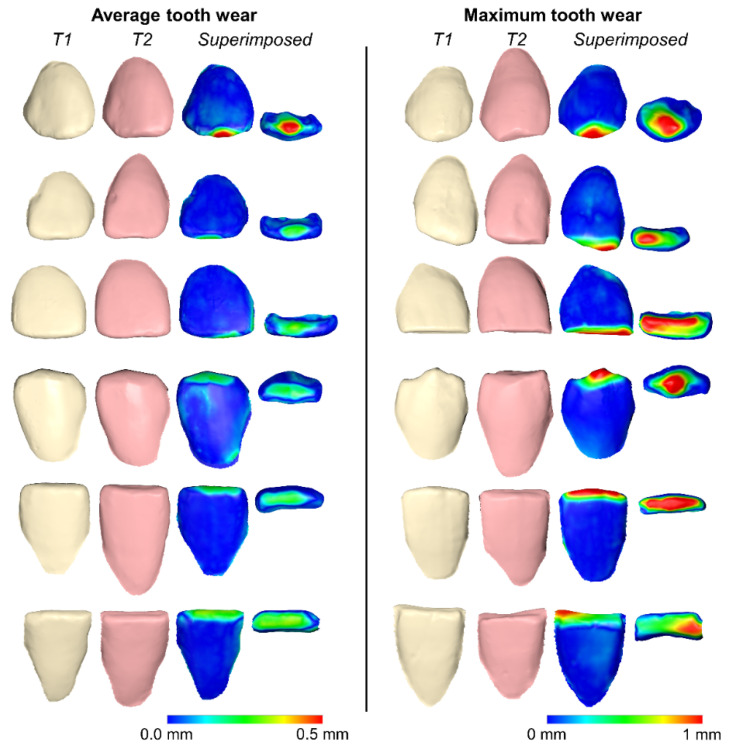
Three-dimensional superimpositions of serial tooth crowns and corresponding color-coded distance maps of representative cases showing the average and the maximum volumetric tooth wears detected per tooth type for the entire sample. From top to bottom: upper canine, upper lateral, upper central, lower canine, lower lateral, and lower central. T1: tooth crowns at end of orthodontic treatment. T2: the same tooth crowns as in T1, on average 13 years after T1.

**Table 1 biology-10-00660-t001:** Teeth that comprised the tooth wear measurement sample.

Canines		Lateral Incisors		Central Incisors	
Upper	Lower	Upper	Lower	Upper	Lower
*n* = 71	*n* = 65	*n* = 67	*n* = 61	*n* = 65	*n* = 56
53F, 18M	50F, 15M	51F, 16M	47F, 14M	51F, 14M	41F, 15M
67R, 68L	60R, 63L	58R, 63L	55R, 51L	54R, 58L	49R, 45L

F: females, M: males, R: right, and L: left.

**Table 2 biology-10-00660-t002:** Occlusal characteristics of the studied sample.

		T1	T2
Angle Class ^1^	Class I	70	61
	Class II	2 (unilateral)	11 (9 unilateral, 2 bilateral)
Overjet ^2^	Normal	72	70
	Increased	0	2
Overbite ^3^	Normal	65	58
	Reduced *	7	14

^1^ Class I: Molar relationship less than ¼ cusp deviation from Class I (in cases of asymmetrical extractions, premolar and canine occlusion was also considered). ^2^ Normal: 1–4 mm and Increased: >4 mm, and ^3^ Normal: 1–4 mm and Reduced: <1 mm. * Only one patient at T2 had a negative overbite. In all other cases, the overbite was between 0 and 1 mm. T1: time point at end of orthodontic treatment. T2: time point on average 13 years after T1.

**Table 3 biology-10-00660-t003:** Results of the ANCOVA testing the effect of tooth type and sex on the detected tooth wear amount after controlling for the patient and duration of assessment period.

Source	df	F	Sig.
Corrected Model	13	16.35	0.000
Intercept	1	0.08	0.782
Patient	1	3.45	0.064
Assessment period	1	11.37	0.001
Tooth type	5	20.76	0.000
Sex	1	74.57	0.000
Tooth type × Sex	5	2.77	0.018

R-Squared = 0.364 (Adjusted R-Squared = 0.342). df: degrees of freedom. F: F-value. Sig.: Significance shown as *p*-values.

## Data Availability

All data presented in this study are available in the article and the Appendix A. The material used to generate the data is available upon reasonable request from the corresponding author.

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
