# Peer review of "Longitudinal 3D Study of Anterior Tooth Wear from Adolescence to Adulthood in Modern Humans"

_biology, 2021, doi:10.3390/biology10070660_

Round 1

Reviewer 1 Report

The article is within the scope of the journal, and I suggest the following revisions to improve its sentence structure and also its comprehension to readers:

  1. Please revise the text to Simple Summary: Tooth wear is the loss of tooth substance by the non-caries processes of attrition, erosion, and dissolution. The incidence of tooth wear is highest in older age patients, particularly due to tooth grinding, chewing, dietary, and tooth brushing habits. However, the average amount of tooth wear among patients has proved to be controversial. The purpose of this clinical study was to use novel 3D imaging techniques to precisely detect the extent of anterior tooth wear over a thirteen-year period, from adolescence to adulthood. This study showed that tooth wear monitoring is important for dentists to guide their patients about how to prevent it, by using mouth guards to prevent grinding, and cutting out harmful habits that lead to tooth wear.

  1. Please revise the text to: Abstract: Tooth wear can easily be observed as a loss of tooth substance, but its precise measurement is problematic. In this study the use of 3D digital models has allowed the precise measurement of tooth wear in the anterior permanent dentition of seventy-two (n=72) human subjects from adolescence to adulthood who received orthodontic treatment. There was an average loss from the surfaces of anterior occlusal tooth structure of 1.58 mm3 per tooth. Tooth surface loss in at least one tooth was higher than 1 mm3 in 93.1% of the human subjects. Tooth wear was highest in males versus females, and in the upper canine teeth versus lower lateral teeth. These results show that the upper canine teeth are more prone to tooth wear, and that males are more likely to suffer from tooth wear. The use of 3D digital models are recommended to be used by dentists to monitor tooth wear in patients, and to help identify tooth wear in vulnerable and high risk populations.

  1. Introduction, please delete the sentence “Currently, the amount of tooth wear occurring in the long-term on a group of individuals remains largely unknown.” There are many studies which have investigated tooth wear so it cannot be “largely unknown.”

  1. Ethical approval, for adolescents, did the parents/guardians provide their approval for teenagers (under the age of 18) to be included in this research? If not the ethical approvals may not be valid.

  1. Sample, for the population of 72 human subjects, how were they selected? Was it randomized? Or because they showed a high risk of tooth wear? Was the sample population representative?

  1. Tooth wear assessment, how was experimenter bias removed? Were the subjects and teeth measurements blinded? How was bias removed?

  1. Method error. Did examiner (KD) collect all the data? If so how can you be sure that the data was not biased and was checked by an independent examiner?

  1. Standard error, why not standard deviation? Standard error is usually much smaller.

  1. Methods, why did you not have any negative or positive controls to ensure your methods were valid?

  1. Results, when you state “tooth wear” did you mean “tooth surface loss”? If so please revise your text.

  1. 3 Tooth wear, use a citation to support your use of 1mm3 as the cut off?

  1. Figures. Acceptable

  1. References. Acceptable

Reviewer 2 Report

The manuscript is very well written. It is very interesting and it adds new informations to the international literature.

Reviewer 3 Report

Dear Authors,

I just read your interesting manuscript. I just have some minor issues that need to be addressed.

  • You write about a 13-year-period all over the manuscript but according to the paragraph 1 on page 3, the minimum duration was only 9 years. I think this needs to be revised/clarified.
  • Why were only "white" patients included? I think both the term and the approach are kind of difficult (politically ...). While it seems to be reasonable for studies on craniofacial growth to include patients of caucasian origin only, I cannot really see the necessity when it comes to tooth wear ...
  • Why was the age restriction 12-24 years? I could understand if the restriction was rather strict in terms of 12-15 or similar but why use an upper limit if the range is that wide anyway?
  • Why does the number of n differ between the tooth categories (Table 1)? Why were the respective patients not excluded as casts had to be of good quality according to the inclusion criteria?
  • It is stated that all patients had acceptable occlusion at both time points and two overbite categories are given in Table 2. In my opinion, however, an essential criteria would be incisal contact (which is not automatically given by a positive overbite value). This should be stated somewhere respectively should teeth without incisal contact be excluded.
